# Coping with label shift via distributionally robust optimisation

**Jingzhao Zhang**
Massachusetts Institute of Technology
jzhzhang@mit.edu

**Aditya Krishna Menon & Andreas Veit & Srinadh Bhojanapalli & Sanjiv Kumar**
Google Research
{adityakmenon, aveit, bsrinadh, sanjivk}@mit.edu

**Suvrit Sra**
Massachusetts Institute of Technology
suvrit@mit.edu

## Abstract

The label shift problem refers to the supervised learning setting where the train and test label distributions do not match. Existing work addressing label shift usually assumes access to an *unlabelled* test sample. This sample may be used to estimate the test label distribution, and to then train a suitably re-weighted classifier. While approaches using this idea have proven effective, their scope is limited as it is not always feasible to access the target domain; further, they require repeated retraining if the model is to be deployed in *multiple* test environments. Can one instead learn a *single* classifier that is robust to arbitrary label shifts from a broad family? In this paper, we answer this question by proposing a model that minimises an objective based on distributionally robust optimisation (DRO). We then design and analyse a gradient descent-proximal mirror ascent algorithm tailored for large-scale problems to optimise the proposed objective. Finally, through experiments on CIFAR-100 and ImageNet, we show that our technique can significantly improve performance over a number of baselines in settings where label shift is present.

## 1 Introduction

Classical supervised learning involves learning a model from a training distribution that generalises well on test samples drawn from the *same* distribution. While the assumption of identical train and test distributions has given rise to useful methods, it is often violated in many practical settings (Kouw & Loog, 2018). The *label shift* problem is one such important setting, wherein the training distribution over the labels does not reflect what is observed during testing (Saerens et al., 2002). For example, consider the problem of object detection in self-driving cars: a model trained in one city may see a vastly different distribution of pedestrians and cars when deployed in a different city. Such shifts in label distribution can significantly degrade model performance. As a concrete example, consider the performance of a ResNet-50 model on ImageNet. While the overall error rate is $\sim 24\%$, Figure 1 reveals that certain classes suffer an error as high as $\sim 80\%$. Consequently, a label shift that increases the prevalence of the more erroneous classes in the test set can significantly degrade performance.

Most existing work on label shift operates in the setting where one has an *unlabelled* test sample that can be used to estimate the shifted label probabilities (du Plessis & Sugiyama, 2014; Lipton et al., 2018; Azizzadenesheli et al., 2019). Subsequently, one can retrain a classifier using these probabilities in place of the training label probabilities. While such techniques have proven effective, it is not always feasible to access an unlabelled set. Further, one may wish to deploy a learned model in *multiple* test environments, each one of which has its own label distribution. For example, the label distribution for a vehicle detection camera may change continuously while driving across the city. Instead of simply deploying a separate model for each scenario, deploying a *single* model that is

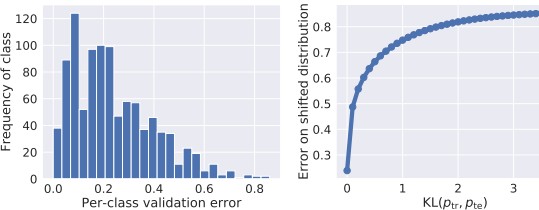

Figure 1: Distribution of per-class test errors of a ResNet-50 on ImageNet (left). While the average error rate is $\sim 24\%$, some classes achieve an error as high as $\sim 80\%$. An adversary can thus significantly degrade test performance (right) by choosing $p_{\text{te}}(y)$ with more weight on these classes.

robust to shifts may be more efficient and practical. Hence, we address the following question in this work: *can we learn a single classifier that is robust to a family of arbitrary shifts?*

We answer the above question by modeling label shift via distributionally robust optimisation (DRO) (Shapiro et al., 2014; Rahimian & Mehrotra, 2019). DRO offers a convenient way of coping with distribution shift, and have lead to successful applications (e.g. Faury et al. (2020)). Intuitively, by seeking a model that performs well on *all* label distributions that are "close" to the training data label distribution, this task can be cast as a game between the learner and an *adversary*, with the latter allowed to pick label distributions that maximise the learner's loss. We remark that while adversarial perspectives have informed popular paradigms such as GANs, these pursue fundamentally different objectives from DRO (see Appendix A for details).

Although several previous works have explored DRO for tackling the problem of *example* shift (e.g., adversarial examples) (Namkoong & Duchi, 2016; 2017; Duchi & Namkoong, 2018), an application of DRO to the label shift setting poses several challenges: (a) updating the adversary's distribution naïvely requires solving a nontrivial convex optimisation subproblem with limited tractability, and also needs careful parameter tuning; and (b) naïvely estimating gradients under the adversarial distribution on a randomly sampled minibatch can lead to unstable behaviour (see §3.1). We overcome these challenges by proposing the first algorithm that successfully optimises a DRO objective for label shift on a large scale dataset (i.e., ImageNet). Our objective encourages robustness to arbitrary label distribution shifts within a *KL-divergence ball* of the empirical label distribution. Importantly, we show that this choice of robustness set admits an *efficient* and *stable* update step.

**Summary of contributions**

(1) We design a gradient descent-proximal mirror ascent algorithm tailored for optimising large-scale problems with minimal computational overhead, and prove its theoretical convergence.

(2) With the proposed algorithm, we implement a practical procedure to successfully optimise the robust objective on ImageNet scale for the label shift application.

(3) We show through experiments on ImageNet and CIFAR-100 that our technique significantly improves over baselines when the label distribution is adversarially varied.

## 2 BACKGROUND AND PROBLEM FORMULATION

In this section we formalise the label shift problem and motivate its formulation as an adversarial optimisation problem. Consider a multiclass classification problem with distribution $p_{\text{tr}}$ over instances $\mathcal{X}$ and labels $\mathcal{Y} = [L]$. The goal is to learn a classifier $h_\theta \colon \mathcal{X} \to \mathcal{Y}$ parameterised by $\theta \in \Theta$, with the aim of ensuring good predictive performance on future samples drawn from $p_{\text{tr}}$. More formally, the goal is to minimise the objective $\min_\theta \mathbb{E}_{(x,y) \sim p_{\text{tr}}}[\ell(x, y, \theta)]$, where $\ell \colon \mathcal{X} \times \mathcal{Y} \times \Theta \to \mathbb{R}_+$ is a loss function. In practice, we only have access to a finite sample $\mathcal{S} = \{(x_i, y_i)\}_{i=1}^n \sim p_{\text{tr}}^n$, which motivates us to use the empirical distribution $p_{\text{emp}}(x, y) = \frac{1}{n} \sum_{i=1}^n \mathbb{1}(x = x_i, y = y_i)$ in place of $p_{\text{tr}}$. Doing so, we arrive at the objective of minimising the empirical risk:

$$\min_\theta \ \mathbb{E}_{p_{\text{emp}}}[\ell(x, y, \theta)] := \frac{1}{n} \sum_{i=1}^n \ell(x_i, y_i, \theta). \tag{1}$$

The assumption underlying the above formulation is that test samples are drawn from the same distribution $p_{\text{tr}}$ that is used during training. However, this assumption is violated in many practical settings. The problem of learning from a training distribution $p_{\text{tr}}$, while attempting to perform well on a test distribution $p_{\text{te}} \neq p_{\text{tr}}$ is referred to as *domain adaptation* (Ben-David et al., 2007). In

| Label distribution | Reference |
|---|---|
| Train distribution | Standard ERM |
| Specified a-priori (e.g., balanced) | (Elkan, 2001; Xie & Manski, 1989; Cao et al., 2019) |
| Estimated test distribution | (du Plessis & Sugiyama, 2014; Lipton et al., 2018; Azizzadenesheli et al., 2019; Garg et al., 2020; Combes et al., 2020) |
| Worst-performing class | (Hashimoto et al., 2018; Mohri et al., 2019; Sagawa et al., 2020) |
| Worst $k$-performing classes | (Fan et al., 2017; Williamson & Menon, 2019; Curi et al., 2019; Duchi et al., 2020) |
| Adversarial shifts within KL-divergence | This paper |

Table 1: Summary of approaches to learning with a modified label distribution.

the special case of *label shift*, one posits that $p_{\text{te}}(x \mid y) = p_{\text{tr}}(x \mid y)$, but the label distribution $p_{\text{te}}(y) \neq p_{\text{tr}}(y)$ (Saerens et al., 2002); i.e., the test distribution satisfies $p_{\text{te}}(x, y) = p_{\text{te}}(y)p_{\text{tr}}(x \mid y)$. The label shift problem admits the following three distinct settings (see Table 1 for a summary):

**(1) Fixed label shift**. Here, one assumes *a-priori* knowledge of $p_{\text{te}}(y)$. One may then adjust the outputs of a probabilistic classifier post-hoc to improve test performance (Elkan, 2001). Even when the precise distribution is unknown, it is common to posit a uniform $p_{\text{te}}(y)$. Minimising the resulting *balanced error* has been the subject of a large body of work (He & Garcia, 2009), with recent developments including Cui et al. (2019); Cao et al. (2019); Kang et al. (2020); Guo et al. (2020).

**(2) Estimated label shift**. Here, we assume that $p_{\text{te}}(y)$ is unknown, but that we have access to an unlabelled test sample. This sample may be used to estimate $p_{\text{te}}(y)$, e.g., via kernel mean-matching (Zhang et al., 2013), minimisation of a suitable KL divergence (du Plessis & Sugiyama, 2014), or using black-box classifier outputs (Lipton et al., 2018; Azizzadenesheli et al., 2019; Garg et al., 2020). One may then use these estimates to minimise a suitably re-weighted empirical risk.

**(3) Adversarial label shift**. Here, we assume that $p_{\text{te}}(y)$ is unknown, and guard against a suitably defined worst-case choice. Observe that an extreme case of label shift involves placing all probability mass on a single $y^* \in \mathcal{Y}$. This choice can be problematic, as (1) may be rewritten as

$$\min_{\theta} \sum_{y \in [L]} p_{\text{emp}}(y) \cdot \left\{ \frac{1}{n_y} \sum_{i : \, y_i = y} \ell(x_i, y_i, \theta) \right\},$$

where $n_y$ is the number of training samples with label $y$. The empirical risk is thus a weighted average of the per-class losses. Observe that if some $y^* \in \mathcal{Y}$ has a large per-class loss, then an adversary could degrade performance by choosing a $p_{\text{te}}$ with $p_{\text{te}}(y^*)$ being large. One means of guarding against such adversarial label shifts is to minimise the *minimax risk* (Alaiz-Rodríguez et al., 2007; Davenport et al., 2010; Hashimoto et al., 2018; Mohri et al., 2019; Sagawa et al., 2020)

$$\min_{\theta} \max_{\pi \in \Delta^L} \sum_{y \in [L]} \pi(y) \cdot \left\{ \frac{1}{n_y} \sum_{i : \, y_i = y} \ell(x_i, y_i, \theta) \right\}, \tag{2}$$

where $\Delta^L$ denotes the simplex. In (2), we combine the per-label risks according to the *worst-case* label distribution. In practice, focusing on the worst-case label distribution may be overly pessimistic. One may temper this by instead constraining the label distribution. A popular choice is to enforce that $\|\pi\|_\infty \leq \frac{1}{k}$ for suitable $k$, which corresponds to minimising the *average of the top-k largest* per-class losses for integer $k$ (Williamson & Menon, 2019; Curi et al., 2019; Duchi et al., 2020).

We focus on the adversarial label shift setting, as it meets the desiderata of training a single model that is robust to multiple label distributions, and not requiring access to test samples. Adversarial robustness has been widely studied (see Appendix A for more related work), but its application to label shift is much less explored. Amongst techniques in this area, Mohri et al. (2019); Sagawa et al. (2020) are most closely related to our work. These works optimise the worst-case loss over subgroups induced by the labels. However, both works consider settings with a relatively small ($\leq 10$) number of subgroups; the resultant algorithms face many challenges when trained with many labels (see Section 4). We now detail how a suitably constrained DRO formulation, coupled with optimisation choices, can overcome this limitation.

---

**Algorithm 1** ADVSHIFT($\theta_0, \gamma_c, \lambda, \texttt{NNOpt}, p_{\text{emp}}, \eta_\pi$)

---

1: Initialise adversary distribution as $\pi_1 = (\frac{1}{L}, ..., \frac{1}{L})$.
2: **for** $t = 1, \ldots, T$ **do**
3:    Sample mini-batch of $b$ examples $\{(x_i, y_i)\}_{i=1}^b$.
4:    Evaluate stochastic gradient $g_\theta = \frac{1}{b} \sum_{i=1}^b \frac{\pi_t(y_i)}{p_{\text{emp}}(y_i)} \cdot \nabla_\theta \ell(x_i, y_i, \theta_t)$
5:    Update neural network parameters $\theta_{t+1} = \texttt{NNOpt}(g_\theta)$
6:    Update Lagrangian variable $\alpha = 0$ if $r > \text{KL}(\pi_t, p_{\text{emp}})$, $\alpha = 2\gamma_c \lambda$ if $r < \text{KL}(\pi_t, p_{\text{emp}})$.
7:    Evaluate adversarial gradient $g_\pi(i) = \frac{1}{b} \sum_{j=1}^b \frac{\mathbb{1}\{y_j = i\}}{p_{\text{emp}}(i)} \cdot \nabla_\pi \ell(x_j, y_j, \theta_{t+1})$.
8:    Update adversarial distribution $\pi_{t+1} = (\pi_t \cdot p_{\text{emp}}^\alpha)^{1/(1+\alpha)} \cdot \exp(\eta_\pi g_\pi)/C$

---

## 3 ALGORITHM: DISTRIBUTIONALLY ROBUST KL-DIVERGENCE MINIMISATION

To address the adversarial label shift problem, we propose to replace the empirical risk (1) with

$$\min_\theta \max_{\pi \in \mathcal{P}} \mathbb{E}_\pi[\ell(x, y, \theta)], \quad \mathcal{P} := \{\pi \in \Delta^L \mid d(\pi, p_{\text{emp}}) \leq r\}, \tag{3}$$

where $\mathcal{P}$ is an *uncertainty set* containing perturbations of the empirical distribution $p_{\text{emp}}$. This is an instance of *distributionally robust optimisation* (*DRO*) (Shapiro et al., 2014), a framework where one minimises the *worst-case* expected loss over a family of distributions. In this work, we instantiate DRO with $\mathcal{P}$ being a parameterised family of distributions with varying marginal label distributions in KL-divergence, i.e., $d(p, q) = \mathbb{E}_{y \sim q}[-\log p(y)/q(y)]$. (We use this divergence, as opposed to a generic $f$-divergence, as it affords closed-form updates; see §3.3.) Solving (3) thus directly addresses adversarial label shift, as it ensures our model performs well on arbitrary label distributions from $\mathcal{P}$. Observe further that the existing minimax risk (2) is a special case of (3) with $r = +\infty$.

Having stated our learning objective, we now turn to the issue of how to optimise it. One natural thought is to leverage strategies pursued in the literature on *example-level DRO* using $f$-divergences. For example, Namkoong & Duchi (2016) propose an algorithm that alternately performs iterative gradient updates for model parameters $\theta$ and adversarial distribution $\pi$, assuming access to projection oracles, and the ability to sample from the adversarial distribution. However, there are challenges in applying such techniques on large-scale problems (e.g., ImageNet):

(1) directly sampling from $\pi$ is challenging in most data loading pipelines for ImageNet.
(2) projecting $\pi$ onto the feasible set $\mathcal{P}$ requires solving a constrained convex optimization problem at every iteration, which can incur non-trivial overhead (see Appendix E).

We now describe ADVSHIFT (Algorithm 1), our approach to solve these problems. In a nutshell, we iteratively update model parameters $\theta$ and adversarial distributions $\pi$. In the former, we update exactly as per ERM optimization (e.g., ADAM, SGD), which we denote as $\texttt{NNOpt}$ (neural network optimiser); in the latter, we introduce a Lagrange multiplier to avoid projection. Extra care is needed to obtain unbiased gradients and speed up adversarial convergence, as we now detail.

### 3.1 ESTIMATING THE ADVERSARIAL MINIBATCH GRADIENT

For a fixed $\pi \in \Delta^L$, to estimate the parameter gradient $\mathbb{E}_\pi[\nabla_\theta \ell(x, y, \theta)]$ on a training sample $\mathcal{S} = \{(x_i, y_i)\}_{i=1}^n$, we employ the importance weighting identity and write

$$\mathbb{E}_\pi[\nabla_\theta \ell(x, y, \theta)] = \mathbb{E}_{p_{\text{emp}}}\left[\frac{1}{p_{\text{emp}}(y)} \cdot \nabla_\theta \ell(x, y, \theta)\right] = \frac{1}{n} \sum_i \frac{\pi(y_i)}{p_{\text{emp}}(y_i)} \cdot \nabla_\theta \ell(x_i, y_i, \theta).$$

We may thus draw a minibatch as usual from $\mathcal{S}$, and apply suitable weighting to obtain unbiased gradient estimates. A similar reweighting is necessary to compute the adversary gradients $\mathbb{E}_\pi[\nabla_\pi \ell(x, y, \theta)]$. Making the adversarial update efficient requires further effort, as we now discuss.

### 3.2 REMOVING CONSTRAINTS BY LAGRANGIAN DUALITY

To efficiently update the adversary distribution $\pi$ in (3), we would like to avoid the cost of projecting onto $\mathcal{P}$. To bypass this difficulty, we make the following observation based on Lagrangian duality.

**Proposition 1.** *Suppose $\ell$ is bounded, and $p_{\mathrm{emp}}$ is not on the boundary of the simplex. Then, $\forall r > 0$, $\exists \gamma^* > 0$ such that for every $\gamma_c \geq \gamma^*$, the constrained objective is solvable in unconstrained form:*

$$\operatorname*{argmax}_{\pi \in \Delta^L, \, \mathrm{KL}(\pi, p_{\mathrm{emp}}) \leq r} \mathbb{E}_\pi[\ell(x, y, \theta)] = \operatorname*{argmax}_{\pi \in \Delta^L} \mathbb{E}_\pi[\ell(x, y, \theta)] + \min\{0, \gamma_c(r - \mathrm{KL}(\pi, p_{\mathrm{emp}}))\}.$$

Motivated by this, we may thus transform the objective (3) into:

$$\min_\theta \max_{\pi \in \Delta^L} \mathbb{E}_\pi[\ell(x, y, \theta)] + \min\{0, \gamma_c(r - \mathrm{KL}(\pi, p_{\mathrm{emp}}))\}, \tag{4}$$

where $\gamma_c > 0$ is a sufficiently large constant; in practice, this may be chosen by a bisection search. The advantage of this formulation is that it admits an efficient update for $\pi$, as we now discuss.

### 3.3 Adversarial distribution updates

We now detail how we can employ *proximal mirror descent* to efficiently update $\pi$. Observe that we may decompose the adversary's (negated) objective into two terms: $f(\theta, \pi) := -\mathbb{E}_\pi[\ell(x, y, \theta)]$ and $h(\pi) := \max\{0, \gamma_c(\mathrm{KL}(\pi, p_{\mathrm{emp}}) - r)\}$, where $h(\pi)$ is independent of the samples. Such decomposable objectives suggest using *proximal* updates (Combettes & Pesquet, 2011):

$$\pi_{t+1} = \operatorname{prox}_{\lambda h}(\pi_t - \lambda \nabla_\pi f(\theta_t, \pi_t)) := \operatorname*{argmin}_{\pi \in \Delta^L} h(\pi) + \frac{1}{2\lambda}(\|\pi_t - \pi\|^2 + 2\lambda \langle \nabla_\pi f(\theta_t, \pi_t), \pi \rangle), \tag{5}$$

where $\lambda$ serves as the learning rate. The value of proximal descent relies on the ability to efficiently solve the minimisation problem in (5). Unfortunately, this does not hold as-is for our choice of $h(\pi)$, essentially due to a mismatch between the use of KL-divergence in $h$, and Euclidean distance $\|\pi_t - \pi\|^2$ in (5). Motivated by the advantages of *mirror descent* over gradient descent on the simplex (Bubeck, 2014), we propose to replace the Euclidean distance with KL-divergence:

$$\pi_{t+1} = \operatorname*{argmin}_{\pi \in \Delta^L} h(\pi) + \frac{1}{2\lambda}(\mathrm{KL}(\pi, \pi_t) + 2\lambda \langle g_t, \pi \rangle), \tag{6}$$

where $g_t$ is an unbiased estimator of $\nabla_\pi f(\theta_t, \pi_t)$. We have the following closed-form update.

**Lemma 2.** *Assume the optimal solution $\pi_{t+1}$ to (6) satisfies $\mathrm{KL}(\pi_{t+1}, p_{\mathrm{emp}}) \neq r$, and that all the classes appeared at least once in the empirical distribution, i.e. $\forall i, p_{\mathrm{emp}}^i > 0$. Let $\gamma = \gamma_c$ if $r < \mathrm{KL}(\pi_{t+1}, p_{\mathrm{emp}})$, and $\gamma = 0$ if $r > \mathrm{KL}(\pi_{t+1}, p_{\mathrm{emp}})$, then $\pi_{t+1}$ permits a closed form solution*

$$\pi_{t+1} = (\pi_t \odot p_{\mathrm{emp}}^\alpha)^{1/(1+\alpha)} \exp(\eta_\pi g_t)/C, \tag{7}$$

*where $\eta_\pi = \frac{1}{(\gamma + 1/2\lambda)(1+\alpha)}$, $\alpha = 2\gamma\lambda$, $C = \|(\pi_t \odot p_{\mathrm{emp}}^\alpha)^{1/(1+\alpha)} \exp(\eta_\pi g_t)\|_1$ projects $\pi_{t+1}$ onto the simplex, and $a \odot b$ is the element-wise product between two vectors $a, b$.*

In Algorithm 1, we set $\gamma = \gamma_c$ if $r < \mathrm{KL}(\pi_t, p_{\mathrm{emp}})$ and 0 otherwise to appoximate the true $\gamma$. Such approximation works well when $r - \mathrm{KL}(\pi_t, p_{\mathrm{emp}})$ does not change sign frequently.

### 3.4 Convergence Analysis

We provide below a convergence analysis of our gradient descent-proximal mirror ascent method for nonconvex-concave stochastic saddle point problems. For the composite objective $\min_\theta \max_{\pi \in \Delta^L} f(\theta, \pi) + h(\pi)$, and fixed learning rate $\eta_\theta$, we abstract the Algorithm 1 update as:

$$\theta_{t+1} = \theta_t - \eta_\theta g(\theta_t), \quad \pi_{t+1} = \operatorname*{argmax}_\pi h(\pi) - \frac{1}{2\lambda}(\mathrm{KL}(\pi, \pi_t) + 2\lambda \langle g(\pi_t), \pi \rangle), \tag{8}$$

where $g(\pi), g(\theta)$ are stochastic gradients assumed to satisfy the following.

**Assumption 1.** The stochastic gradient $g(\theta)$ with respect to $\theta$ satisfies that for some $\sigma > 0$,

$$\mathbb{E}[g(\theta)] = \nabla_\theta f(\theta, \pi), \text{ and } \mathbb{E}[\|g(\theta) - \mathbb{E}[g(\theta)]\|^2] \leq \sigma^2.$$

**Assumption 2.** The stochastic gradient $g(\pi)$ with respect to $\pi$ satisfies that for some $G > 0$,

$$\mathbb{E}[g(\pi)] = \nabla_\pi f(\theta, \pi), \text{ and } \mathbb{E}[\|g(\pi)\|_\infty^2] \leq G^2.$$

We make the following assumptions about the objective, similar to Lin et al. (2019; 2020):

**Assumption 3.** $f(\theta, \pi) + h(\pi)$ is $L-$smooth and $l-$Lipschitz; $f(\theta, \pi)$ and $h(\pi)$ are concave in $\pi$.

**Assumption 4.** Every adversarial distribution iterate $\pi_t$ satisfies $\mathrm{KL}(\pi_t, p_{\mathrm{emp}}) \leq R$ for some $R > 0$.

Assumption 3 and 4 may be enforced by adding a constant $\epsilon$ to the adversarial updates, which prevents $\pi_t$ from approaching the boundary of the simplex. Assumption 2 in the label shift setting implies that the loss is upper and lower bounded. Such an assumption may be enforced by clipping the loss for computing the adversarial gradient, which can significantly speed up training (see Appendix **??**). Furthermore, this is a standard assumption for analyzing nonconvex-concave problems (Lin et al., 2019). The assumption that the square $L_\infty$ norm is bounded is weaker than $L_2$ norm being bounded; such a relaxation results from using mirror rather than Euclidean update.

Given that the function $F(\theta) := \max_{\pi \in \Delta} f(\theta, \pi) + h(\pi)$ is nonconvex, our goal is to find a stationary point instead of approximating global optimum. Yet, due to the minimax formulation, the function $F(\theta)$ may not necessarily be differentiable. Hence, we define convergence following some recent works (Davis & Drusvyatskiy, 2019; Lin et al., 2019; Thekumparampil et al., 2019) on nonconvex-concave optimisation. First, Assumption 3 implies $F(\theta)$ is $L-$weakly convex and $l$-Lipschitz (Lin et al., 2019, Lemma 4.7). Hence, we define stationarity in the language of weakly convex functions.

**Definition 1.** A point $\theta$ is an $\epsilon-$stationary point of a weakly convex function $F$ if $\|\nabla F_{1/2L}(\theta)\| \leq \epsilon$, where $F_{1/2L}(\theta)$ denotes the Moreau envelope $F_{1/2L}(\theta) = \min_w F(w) + L\|w - \theta\|^2$.

With the above definition, we can establish convergence of the following update:

**Theorem 3** (informal). *Under Assumptions 1–4, the update in (8) finds a point $\theta$ with $\mathbb{E}[\|\nabla F_{\frac{1}{2L}}(\theta)\|] \leq \epsilon$ in $\mathcal{O}(\epsilon^{-8})$ iterations.*

For a precise description of the theorem, please see Appendix H. The above result matches the best known rate in Lin et al. (2019) for optimising nonconvex-concave problem with stochastic gradients. To our knowledge, this is the first result that studies convergence of composite objectives with proximal methods under nonconvex-concave settings. By utilizing the proximal operator, it solves the objective with an extra $h(\pi)$ term without incurring additional complexity cost.

### 3.5 CLIPPING AND REGULARISING FOR FASTER CONVERGENCE

In addition to the proposed algorithm, we apply two additional techniques. We explain them here with motivations. First, we also observe that the adversarial's update could be very sensitive to the adversarial gradient $g_k$, i.e. label-wise loss in each minibatch, because the gradient appears in the exponential of the update. To avoid convergence degradation resulted from the noise in $g_k$, we clip the label-wise loss at value 2. Second, we notice that the KL divergence from any interior point of a simplex to its boundary is infinity. Hence, updates near boundary can be highly unstable due to the nonsmooth KL loss. To cope with this, we add a constant $\epsilon$ term on the adversarial distribution to avoid the adversarial distribution reaching any of the vertices on the simplex. The $\epsilon$ term and clipping is critical in both training and convergence analysis. We conduct an ablation of the sensitivity to these parameters in Figures 5 and 6. Note that the experiments show that even without these tricks, our proposed algorithm alone still outperform baselines.

### 3.6 DISCUSSION AND COMPARISON TO EXISTING ALGORITHMS

A number of existing learning paradigms (e.g., fairness, adversarial training, and domain adaptation) have connections to the problem of adversarial label shift; see Appendix A for details.

We comment on some key differences between ADVSHIFT and related techniques in the literature. For the problem of minimising the worst-case loss (2) — which is equivalent to setting the radius $r = +\infty$ in (3) — Sagawa et al. (2020) propose an algorithm that assumes the ability to sample data from a given group in order to evaluate adversarial gradients. Such sampling is cumbersome to implement in most ImageNet data loading pipelines. Mohri et al. (2019) propose a way to evaluate gradients using importance sampling, and then apply projected gradient descent-ascent. This method suffers from instability owing to sampling (upon which we improve with proximal updates), and incurs a non-trivial computational overhead due to the projection step. We will illustrate these

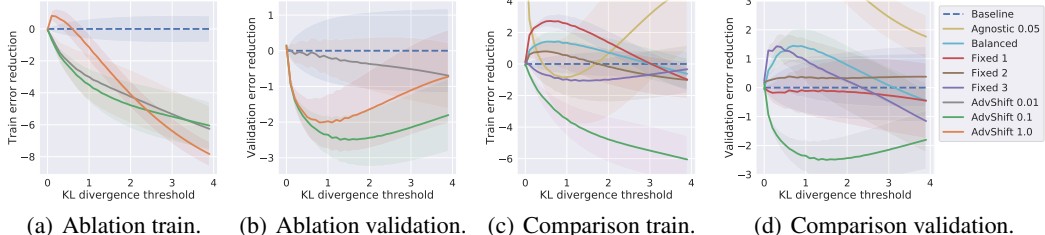

(a) Ablation train.  (b) Ablation validation.  (c) Comparison train.  (d) Comparison validation.

Figure 2: Comparison of performance on ImageNet under adversarial label distributions. For each method, we vary the KL divergence threshold $\tau$, and for each $\tau$ report the maximal validation error induced by the adversarial shift within the threshold. Subplots (a) (b) compare the performance of ADVSHIFT trained with different DRO radius $r$ against the default ERM training. We subtract the baseline error of ERM from all values for easy visualization. Absolute values can be found in Figure 8 in the Appendix. Combined with (c), (d), we see that ADVSHIFT can reduce the adversarial validation error by over $\sim 2.5\%$ compared to the BASELINE method and is consistently superior to the AGNOSTIC, BALANCED and FIXED methods. Figure 3(c) illustrates adversarial distributions for varying thresholds $\tau$.

problems in our subsequent experiments (see results for AGNOSTIC in §4). Finally, for an uncertainty set $\mathcal{P}$ based on the CVaR, Curi et al. (2019) provide an algorithm that updates weights using EXP3. This approach relies on a determinantal point process, which has a poor dimension-dependence.

## 4 EXPERIMENTAL RESULTS

We now present a series of experiments to evaluate the performance of the proposed ADVSHIFT algorithm and how it compares to related approaches from the literature. We first explain our experiment setups and evaluation methods. We then present the results on ImageNet dataset, and show that under the adversarial validation setting, our proposed algorithm significantly outperforms other methods discussed in Table 1. Similar results on CIFAR-100 are shown in the Appendix.

### 4.1 EXPERIMENTAL SETUP

To evaluate the proposed method, we use the standard image classification setup of training a ResNet-50 on ImageNet using SGD with momentum as the neural network optimiser. All algorithms are run for 90 epochs, and are found to take almost the same clock time. Note that ImageNet has a largely balanced training label distributions, and perfectly balanced validation label distributions.

We assess the performance of models under adversarial label shift as follows. First, we train a model on the training set and compute its error distribution on the validation set. Next, we pick a threshold $\tau$ on the allowable KL divergence between the train and target distribution and find the adversarial distribution within this threshold which achieves the worst-possible validation error. Finally, we compute the validation performance under this distribution. Note that $\tau = 0$ corresponds to the train distribution, while $\tau = +\infty$ corresponds to the worst-case label distribution (see Figure 1).

We evaluate the following methods, each corresponding to one row in Table 1: (i) standard empirical risk minimisation (BASELINE) (ii) balanced empirical risk minimisation (BALANCED) (iii) agnostic federated learning algorithm of Mohri et al. (2019), which minimises the worst-case loss (AGNOSTIC) (iv) our proposed KL-divergence based algorithm, for various choices of adversarial radius $r$ (AD-VSHIFT) (v) training with ADVSHIFT with a *fixed* adversarial distribution extracted from Figure 3(c) (FIXED). This corresponds to the estimated test distribution row in Table 1 with an ideal estimator.

### 4.2 RESULTS AND DISCUSSION

Figure 2 shows the train and validation performance on ImageNet. Each curve represents the average and standard deviation across 10 independent trials. To better illustrate the differences amongst methods, we plot the difference in error to the BASELINE method. (See Figure 8 in the Appendix for unnormalised plots.) Subfigures (a) and (b) compre the performance of ADVSHIFT for various choices of radius $r$ to the ERM baseline; (c) and (d) compare ADVSHIFT to the remaining methods.

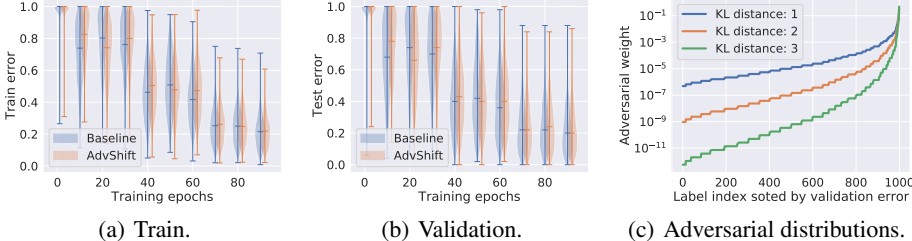

(a) Train.        (b) Validation.        (c) Adversarial distributions.

Figure 3: Subplots (a) (b) show violin plots of the distribution of errors for both the BASELINE and our ADVSHIFT methods over the course of training. On the training set, ADVSHIFT significantly reduces the worst-case error, evidenced by lower upper endpoints of the distribution. On the validation set, the reduction is consistent, albeit less pronounced owing to a generalisation gap. Subplot (c) illustrates adversarial distributions at KL distances of 1, 2 and 3 for model trained with BASELINE. Even at $\tau = 1$, the adversarial distribution is highly concentrated on only a few hard labels.

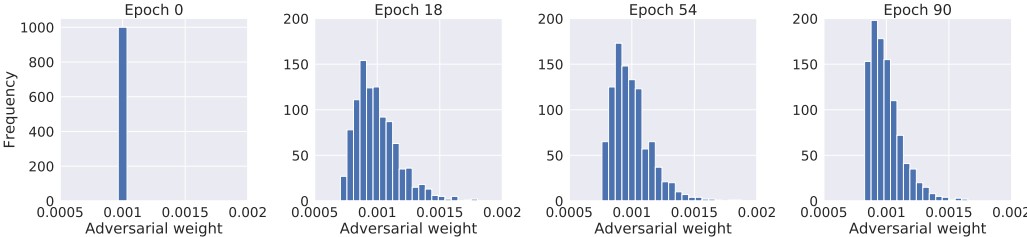

Figure 4: Evolution of learned adversarial distribution ($\pi$) across training epochs. Starting off from a uniform distribution over labels, the adversary quickly infers the relative difficulty of a small fraction of labels, assigning nearly $2\times$ the weight on them compared to the average. This distribution remains largely stable in subsequent iterations, getting gradually more concentrated as training converges.

Hyperparameters for each method are separately tuned. FIXED 1, 2, 3 corresponds to training with each of the three adversarial distributions in Figure 3(c). We see that:

- the reduction offered by ADVSHIFT is consistently superior to that afforded by the AGNOSTIC, BALANCED and FIXED methods. On the training set, we observe significant ($\sim 8\%$) reduction in performance for large KL divergence thresholds. On the validation set, the gains are less pronounced ($\sim 2.5\%$), indicating some degradation due to a generalisation gap.

- while ADVSHIFT consistently improves above the baseline across adversarial radii, we observe best performance for $r = 0.1$. Smaller values of $r$ lead to smaller improvements, while training becomes increasingly unstable for larger radii. Please see the discussion in the last section.

- during training, AGNOSTIC either learns the adversarial distribution too slowly (such that it behaves like ERM), or uses too large a learning rate for the adversary (such that the training fails). This highlights the importance of the proximal mirror ascent updates in our algorithm.

**Illustration of distributions at fixed KL thresholds**. Figure 3(c) visualises the adversarial distributions corresponding to a few values of the KL threshold $\tau$. At a threshold of $\tau = 3$, the adversarial distribution is concentrated on only a few hard labels. Consequently, the resulting performance on such distributions is highly reflective of the worst-case distribution that can happen in reality.

**Training with a fixed adversarial distribution**. Suppose we take the final adversarial distributions shown in Figure 3(c), and then employ them as fixed distributions during training; this corresponds to the *specified a-priori and estimated validation distribution* approaches in Table 1. Does the resulting model similarly reduce the error on hard classes? Surprisingly, Figure 2(d) indicates this is not so, and performance is in fact significantly worse on the "easy" classes. Employing a fixed adversarial distribution may thus lead to underfitting, which has an intuitive explanation: the model must struggle to fit difficult patterns from early stages of training. Similar issues with importance weighting in conjunction with neural networks have been reported in Byrd & Lipton (2019).

**Evolution of error distributions**. To dissect the evolution of performance during training, Figure 3 shows violin plots of the distribution of errors for both the BASELINE and our ADVSHIFT methods

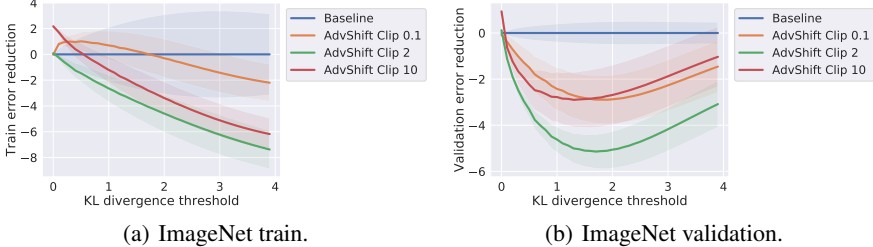

(a) ImageNet train.                    (b) ImageNet validation.

Figure 5: Ablation of loss clipping threshold. We see that when the clipping threshold is either too large or too small, validation performance of the model tends to suffer.

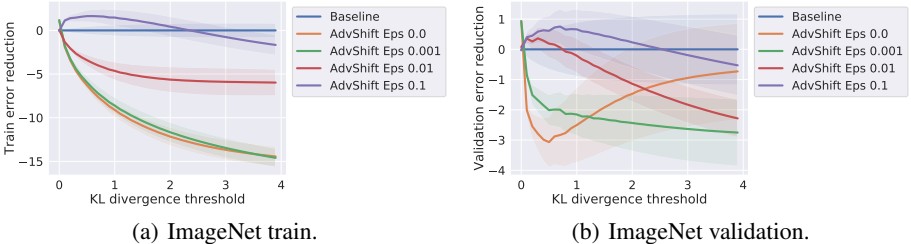

(a) ImageNet train.                    (b) ImageNet validation.

Figure 6: Ablation of gradient stabilisation parameter $\epsilon$, which is a constant added to the gradient updates to prevent iterates from reaching the vertices of the simplex. We see that without any gradient stabilisation, the model's performance rapidly degrades as the adversarial radius increases. Conversely, performance also suffers when the stablisation is too high.

after fixed training epochs. We observe that on the training set, ADVSHIFT significantly reduces the worst-case error, evidenced by the upper endpoints of the distribution being reduced. Note also that, as expected, the adversarial algorithm is slower to reduce the error on the "easy" classes early in training, evidenced by the lower endpoints of the distribution initially taking higher values. On the validation set, the reduction is consistent, albeit less pronounced owing to a generalisation gap.

**Evolution of learned adversarial weights**. To understand the evolution of the adversarial distribution across training epochs, Figure 4 plots the histogram of adversary weights at fixed training epochs. Starting off from a uniform distribution, the adversary is seen to quickly infer the relative difficulty of a small fraction of labels, assigning $\sim 2\times$ the weight on them compared to the average. In subsequent iterations the distribution becomes more concentrated, and gradually reduces the largest weights.

**Ablation of clipping threshold and gradient stabiliser**.

Figures 5 and 6 show an ablation of the choice of loss clipping threshold, and the gradient stabiliser $\epsilon$. We see that when the clipping threshold is either too large or too small, validation performance of the model tends to suffer (albeit still better than the baseline). Similarly, we see that without any gradient stabilisation, the model's performance rapidly degrades as the adversarial radius increases. Conversely, performance also suffers when the stablisation is too high.

In summary, our experiments show that our proposed DRO formulation can be effectively solved with ADVSHIFT, and results in a model that is robust to adversarial label shift.

## 5 DISCUSSION AND FUTURE WORK

We proposed ADVSHIFT, an algorithm for coping with label shift based on distributionally robust optimisation, and illustrated its effectiveness of real-world datasets. Despite this, our approach does not solve the problem fully. First, Figure 2(a)(b) shows that the generalization gap increases as the perturbation radius increases. Understanding why there is a correlation between hard examples and bad generalization could improve robustness. Second, Figure 2(a) shows that even on the train set, the algorithm threshold $r$ does not translate to the model's level of robustness. We conjecture this results from the interplay of model expressivity and data distribution, whose future study is of interest.

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

## A    RELATED PROBLEMS

**Example-level DRO**. Existing work on DRO has largely focussed on the setting where $\mathcal{P}$ encompasses shifts in the instance space (Namkoong & Duchi, 2016; 2017; Sinha et al., 2018; Duchi & Namkoong, 2018; Levy et al., 2020). This notion of robustness has a natural link with adversarial training (Sinha et al., 2017), and involves a more challenging problem, as it requires parameterising the adversary's distribution. Hu et al. (2018) illustrate the potential pitfalls of DRO, owing to a mismatch between surrogate and 0-1 losses. They also propose to encode an uncertainty set based on *latent* label distribution shift (Storkey & Sugiyama, 2007), which requires domain knowledge. The techniques in example-level DRO are mostly designed for small scale dataset with SVM models, as these techniques require sampling according to adversarial distribution, which can be very unstable if implemented with importance sampling only. It also requires maintaining a vector proportional to the number of labels and indexing each sample during training to match up the sample index, which is not available in most dataloading pipelines.

**Fairness**. Adversarial label shift may be related to *algorithmic fairness*. Abstractly, this concerns the mitigation of systematic bias in predictions on sensitive subgroups (e.g., country of origin). One fairness criteria posits that the per-subgroup errors should be equal (Zafar et al., 2017; Donini et al., 2018), an ideal that may be targetted by minimising the worst-subgroup error (Mohri et al., 2019; Sagawa et al., 2020). When the subgroups correspond to labels, ensuring this notion of fairness is tantamount to guarding against an adversary that can place all mass on the worst performing label.

**GANs**. GANs (Goodfellow et al., 2014) involve solving a min-max objective that bears some similarity to the DRO formulation (3), but is fundamentally different in details: while DRO considers reweighting of samples according to a fixed family, GANs involve a parameterised adversarial family, with the training objective augmented with an additional penalty.

**Domain adaptation**. Label shift can be viewed as a special case of domain adaptation, where $p_{tr}$ and $p_{te}$ can systematically differ. Typically, one assumes access to a small sample from $p_{te}$, which may be used to estimate importance weights (Combes et al., 2020), or samples from *multiple* domains, which may be used to estimate a generic domain-agnostic representation (Muandet et al., 2013). In causal inference, there has been interest in similar classes of models (Arjovsky et al., 2019).

## B    ALGORITHM IMPLEMENTATION DETAILS

We introduce some additional details in our implementation of ADVSHIFT. First, as observed in Section 3.1, our algorithm requires knowing the empirical label distribution. As the exact value is not always available, we estimate the empirical label distribution online for all the experiments presented later in Section 4 using an exponential moving average, $p_{emp} = \beta \cdot p_{emp} + (1 - \beta) \cdot p_{batch}$, where $p_{batch}$ is the label distribution in the minibatch. We set $\beta = 0.999$. The number is set such that the exponential moving average has a half-life roughly equal to the number of iterations in one epoch of ImageNet training using our setup.

In all the experiments, we set $2\gamma_c\lambda = 1$ in Algorithm 1 for simplicity. For learning the adversarial distribution, we only tune the adversarial learning rate $\eta_\pi$.

## C    ADDITIONAL EXPERIMENTAL RESULTS

We present here additional experimental results, including:

- for ImageNet, an illustration of the lack of correlation between a label's frequency in the training set, and its validation error. (Figure 7)
- unnormalised versions of the results on ImageNet shown in the body, where we do not subtract the baseline performance from each of the curves; this gives a sense of the absolute performance numbers obtained by each method. (Figure 8)
- an ablation of the loss clipping threshold and gradient stabiliser $\epsilon$ as introduced above. (Figure 5,6)
- results on CIFAR-100, to complement those for ImageNet. (Figure 9,10)

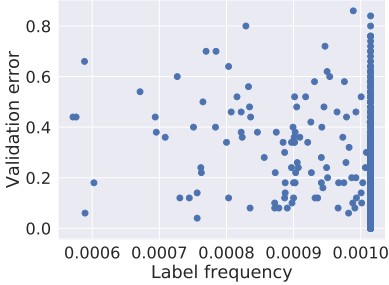

Figure 7: Illustration that training label frequency does not strongly correlate with test error. Observe that several classes with a high error appear frequently in the training set.

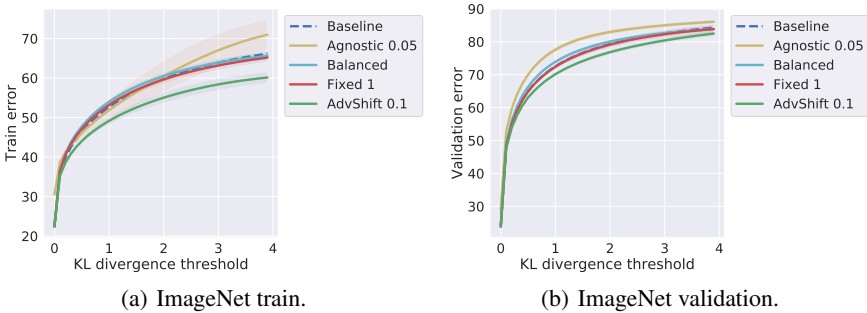

(a) ImageNet train.

(b) ImageNet validation.

Figure 8: Comparison of performance of various methods on ImageNet under adversarial label distributions. For each plot, we vary a KL divergence threshold $\tau$, and for a given $\tau$ construct the label distribution which results in maximal test error for the baseline model. We then compute the test error under this distribution. Note that the case $\tau = 0$ corresponds to using the train distribution, while $\tau = +\infty$ corresponds to using the worst-case label distribution, which is concentrated on the worst-performing label. Our proposed ADVSHIFT can reduce the adversarial test error by over $\sim 2.5\%$ over the baseline method.

## C.1 BALANCED LABELS $\not\Rightarrow$ BALANCED PERFORMANCE

Figure 7 shows that training label frequency does not strongly correlate with test error. Observe that several classes with a high error appear frequently in the training set. Indeed, the three classes with highest error – `casette player`, `maillot`, and `water jug` – all appear an equal number of times in the training set.

## C.2 UNNORMALISED PLOTS ON IMAGENET

Figure 8 presents plots of the unnormalised performance of the various methods compared in the body. Here, rather than subtract the performance of the baseline, we show the absolute accuracy of each method as the adversarial radius is varied. Evidently, the baseline and AGNOSTIC models tend to suffer in their validation error as the adversarial radius increases.

## C.3 RESULTS ON CIFAR-100

Figure 9 shows results on CIFAR-100, where we train various methods using a CIFAR-ResNet-18 as the underlying architecture, Here, we see a consistent and sizable improvement from ADVSHIFT over the baseline method. On this dataset, AGNOSTIC fares better, and eventually matches the performance of ADVSHIFT with a large adversarial radius. This is in keeping with the intended use-case of AGNOSTIC, i.e., minimising the worst-case loss. Figure 10 supplements these plots with unnormalised versions, to illustrate the absolute performance differences.

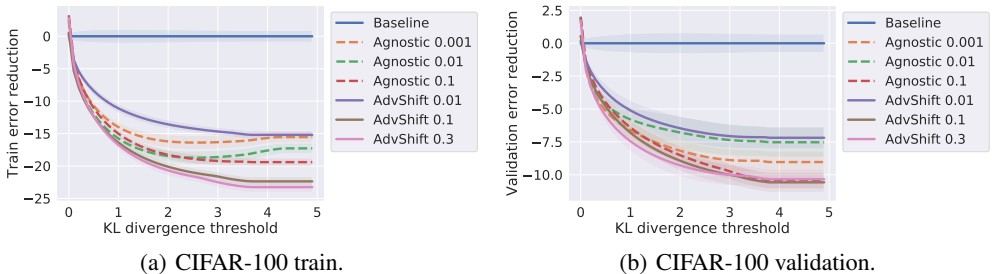

(a) CIFAR-100 train.

(b) CIFAR-100 validation.

Figure 9: Comparison of performance of various methods on CIFAR-100.

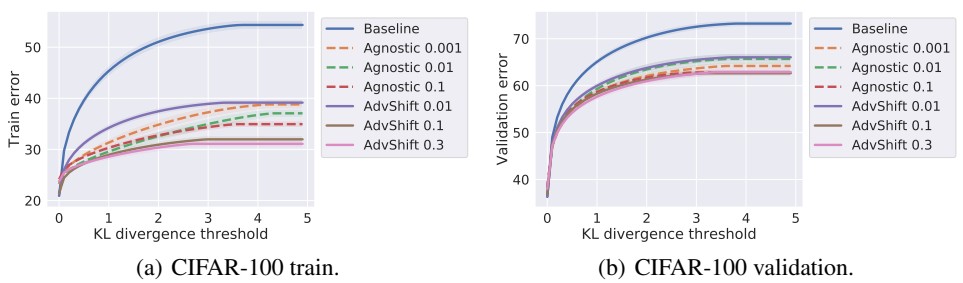

(a) CIFAR-100 train.

(b) CIFAR-100 validation.

Figure 10: Comparison of performance of various methods on CIFAR-100 (unnormalised).

We remark here that the choice of a CIFAR-ResNet-18 results in an underparameterised model, which does not perfectly fit the training data. In the overparameterised case, there are challenges with employing DRO, as noted by Sagawa et al. (2020). Addressing these challenges in settings where the training data is balanced remains an interesting open question.

## D CONSTRAINED DRO DOES NOT PERMIT A BOLTZMAN SOLUTION

We start with a simple example with three label classes $\{a, b, c\}$ with class losses $l = \{1, 2, 4\}$ respectively. We assume an uniform empirical distribution, i.e. $p_{\text{emp}} = \{1/3, 1/3, 1/3\}$. We consider two different problems. The first is to find the optimal solution to regularised objective,

$$p = \operatorname{argmin}_p p^\top l + \gamma \mathrm{KL}(p, p_{\text{emp}}).$$

This problem is well known (e.g. see 2.7.2 of lecture ) to permit a solution of form $p(x) = \frac{\exp l_x/t}{\sum_{x' \in \{a,b,c\}} \exp l'_x/t}$ for some $t$.

In contrast, we show that distributions of the above form does not solve the constrained version of the problem. In particular, we consider the following optimisation problem:

$$\max_p p^\top l$$

$$\text{such that } KL(p, p_{\text{emp}}) \leq r$$

If the solution is of form $p(x) = \frac{\exp l_x/t}{\sum_{x' \in \{a,b,c\}} \exp l'_x/t}$, then we know for $l_b \neq l_c$,

$$(\log(p_a) - \log(p_c))/(\log(p_b) - \log(p_c)) = (l_a - l_c)/(l_b - l_c).$$

We solve the above problem with a convex optimizer using $r = 0.01$ and found $p_a = 0.283, p_b = 0.322, p_c = 0.395$.

$$(\log(p_a) - \log(p_c))/(\log(p_b) - \log(p_c)) = 1.64 \neq (l_a - l_c)/(l_b - l_c) = 1.5.$$

Note that the above example shows that not all solutions of the contrained problem can be written a Boltzman distribution, i.e. $p(x) = \frac{\exp l_x/t}{\sum_{x' \in \{a,b,c\}} \exp l'_x/t}$. Yet, this does not contradict with results

[e.g. Lemma 4, Faury et al. (2020)] that claim there is a Boltzman distribution whose function value matches the optimal value of the constraint problem. Mathematically, we can have $p \neq p'$ but $\mathbb{E}_p[l(x)] = \mathbb{E}_{p'}[l(x)]$.

## E   PROJECTING AN ADVERSARIAL DISTRIBUTION

The projection operator in our setting aims to project a distribution $p$ into the set $\mathcal{P} = \{q : \mathrm{KL}(q, \hat{p}) \leq r\}$ by solving the following problem:

$$\min_q \|q - p\|^2$$

$$\text{such that } \sum_i q_i \log(q_i/p_i) \leq r$$

$$\sum_i q_i = 1,$$

$$\forall i, q_i \geq 0,$$

where $q_i, p_i$ denotes the $i_{th}$ component of $q, p$ and $n$ denotes number of classes. Given that our implementation is based on Tensorflow, we use the "trust-region constrained algorithm" provided by SciPy for easy integration with our python-based training procedure. However, even after extensive tuning, solving each problem up to $1\%$ relative constraint error requires more than 1 minute when $n = 1000$ (the number of labels in ImageNet). This means that if we train ResNet50 on ImageNet for 100k iterations, we need to spend $100k$ minutes on projection operation, which is not affordable.

## F   PROOF OF PROPOSITION 1

*Proof.* We only need to show that for large enough $\gamma_c$, any minimiser $p^*$ of the unconstrained problem satisfies that $\mathrm{KL}(p^*, p_{\mathrm{emp}}) \leq r$. Since the distance from boundary of the simplex to any interior point is $+\infty$, we can safely assume that the point lies within the relative interior of the simplex. To prove the proposition, denote $c = \inf\{\|\nabla_p \mathrm{KL}(p, p_{\mathrm{emp}})\| \mid p \in \Delta^L, s.t. \mathrm{KL}(p, p_{\mathrm{emp}}) > r\}$. By strict convexity of KL divergence and the fact that $r > 0$, we know $c > 0$. Denote the upper bound of loss as $M$, then when $\gamma_c > M/c$, we know that $\mathrm{KL}(p, p_{\mathrm{emp}}) > r \implies 0 \notin \partial_\pi(\mathbb{E}_\pi[\ell(x, y, \theta)] + \min\{0, \gamma_c(r - \mathrm{KL}(\pi, p_{\mathrm{emp}}))\})$. However, since $p$ minimises the objective if and only if $0 \in \partial_\pi(\mathbb{E}_\pi[\ell(x, y, \theta)] + \min\{0, \gamma_c(r - \mathrm{KL}(\pi, p_{\mathrm{emp}}))\})$, we have that any minimiser $p$ must satisfy $\mathrm{KL}(\pi, p_{\mathrm{emp}})) \leq r$. □

## G   PROOF OF LEMMA 2

Recall that we want to find $\pi_{k+1}$ that minimises the following objective,

$$\pi_{k+1} = \underset{\pi \in \Delta}{\mathrm{argmin}}\, h(\pi) + \frac{1}{2\lambda}(\mathrm{KL}(\pi, \pi_k) + 2\lambda\langle g_k, \pi\rangle)$$

$$= \underset{\pi \in \Delta}{\mathrm{argmin}}\, \max\left\{0, \frac{\alpha_c}{1 + \alpha_c}\mathrm{KL}(\pi, p_{\mathrm{emp}})\right\} + \frac{1}{1 + \alpha_c}(\mathrm{KL}(\pi, \pi_k) - \eta\langle g_k, \pi\rangle),$$

where $\eta = 1/(2\gamma_c + 1/\lambda)$, $\alpha_c = 2\gamma_c\lambda$. Denote $v(i)$ as the $i_{th}$ component of vector $v$. Notice that the simplex can be written as a constraint that $\sum_i \pi(i) = 1; \forall i, \pi(i) \geq 0$. Based on this constraint, we first write (6)'s Lagrangian dual

$$L(a, b, \pi) = \sum_i(a_i \pi(i)) + b(\sum_i \pi(i) - 1) + \max\left\{0, \frac{\alpha_c}{1 + \alpha_c}(\mathrm{KL}(\pi, p_{\mathrm{emp}}) - r)\right\}$$

$$+ \frac{1}{1 + \alpha_c}(\mathrm{KL}(\pi, \pi_k) - \eta\langle g_k, \pi\rangle)$$

where $\pi(i)$ denotes the $i_{th}$ component of $\pi$. If $\pi_k > 0$ component-wise, then the optimal $\pi$ cannot lie on the boundary (i.e. $\forall i, \pi(i) > 0$), which results in $\mathrm{KL}(\pi, \pi_k) = \infty$. By Lagrangian duality and

complementary slackness, we know that for if $\mathrm{KL}(\pi, p_{\mathrm{emp}}) > r$,

$$0 = \frac{\partial}{\partial \pi(i)} L(a, b, \pi) = b + \frac{\alpha_c}{1 + \alpha_c} \log(\pi(i)/p_{\mathrm{emp}}(i)) + \frac{1}{1 + \alpha_c} \log(\pi(i)/\pi_k(i)) - \eta g_k(i) + 1.$$

On the other hand, if $\mathrm{KL}(\pi, p_{\mathrm{emp}}) < r$

$$0 = \frac{\partial}{\partial \pi(i)} L(a, b, \pi) = b + \frac{1}{1 + \alpha_c} \log(\pi(i)/\pi_k(i)) - \eta g_k(i) + 1.$$

We discuss the case when $\mathrm{KL}(\pi, p_{\mathrm{emp}}) < r$, and the other case follows similarly. Rearrange the optimality condition of Lagrangian multiplier, we get

$$\frac{\alpha}{1 + \alpha} \log(\pi(i)/p_{\mathrm{emp}}(i)) + \frac{1}{1 + \alpha_c} \log(\pi(i)/\pi_k(i)) - \eta g_k(i) = -b - 1.$$

Since $b$ is a constant for all coordinates,

$$\pi(i) \propto (p_{\mathrm{emp}}(i)_c^\alpha \pi_k(i))^{1/(1+\alpha_c)} \exp\left(\frac{\eta g_k(i)}{1+\alpha_c}\right).$$

The result follows by noting that $\sum_i \pi(i) = 1$.

## H    PROOF OF THEOREM 3

For completeness, we define several terms used in optimisation. A function $f(\theta)$ is $l-$**Lipschitz** if for all $\theta, \theta'$,

$$|f(\theta) - f(\theta')| \leq l\|\theta - \theta'\|.$$

A function $f(\theta)$ is $L-$**smooth** if for all $\theta, \theta'$,

$$\|\nabla f(\theta) - \nabla f(\theta')\| \leq L\|\theta - \theta'\|.$$

A function $f(\theta)$ is $L-$**weakly convex** if $f(\theta) + \frac{L}{2}\|\theta\|^2$ is convex.

Then we can state the formal theorem below.

**Theorem 4** (formal version of Theorem 3). *Under Assumptions 1–4, the update in (8) generates a sequence of points $\theta_1, ..., \theta_T$ with the following property:*

$$\frac{1}{T} \sum_t \mathbb{E}[\|\nabla F_{1/2L}(\theta_t)\|^2] \leq \frac{1}{T^{1/4}} \left( \frac{2}{L}(F_{1/2L}(\theta_0) - F^*_{1/2L}) + 2G + G^2 + \frac{R}{2} + (l^2 + \sigma^2)^{1/2}) \right)$$
$$+ (h^* - h(\pi_0))/T$$

*Proof.* For convenience, denote $\Phi(\theta, \pi) = f(\theta, \pi) + h(\pi)$, $F(\theta) = \max_p \Phi(\theta, p)$. We start by following the standard SGD proof. Denote $g_\theta$ as the stochastic gradient evaluated at step $t - 1$ with respect to $\theta$. Denote $\hat{\theta} = \mathrm{prox}_{F/2L}(\theta) := \arg\min_w \{F(w) + 2L\|w - \theta\|^2\}$. Conditioned on $\theta_{t-1}$, we have

$$\mathbb{E}[\|\hat{\theta}_{t-1} - \theta_t\|^2] = \|\theta_{t-1} - \hat{\theta}_{t-1}\|^2 + 2\eta_\theta \mathbb{E}[\langle \hat{\theta}_{t-1} - \theta_{t-1}, g_\theta \rangle] + \eta_\theta^2 \mathbb{E}[\|g_\theta\|^2]$$
$$\leq \|\theta_{t-1} - \hat{\theta}_{t-1}\|^2 + 2\eta_\theta \langle \hat{\theta}_{t-1} - \theta_{t-1}, \nabla_\theta \Phi(\theta_{t-1}, \pi_{t-1}) \rangle + \eta_\theta^2(L^2 + \sigma^2) \quad (9)$$

where the first equality follows by $\theta_t = \theta_{t-1} - \eta_\theta \nabla_\theta \Phi(\theta_{t-1}, \pi_{t-1})$. Next, we observe that

$$\langle \hat{\theta}_{t-1} - \theta_{t-1}, \nabla_\theta \Phi(\theta_{t-1}, \pi_{t-1}) \rangle \leq \Phi(\hat{\theta}_{t-1}, \pi_{t-1}) - \Phi(\theta_{t-1}, \pi_{t-1}) + \frac{L}{2}\|\hat{\theta}_{t-1} - \theta_{t-1}\|^2 \quad (10)$$
$$\leq F(\hat{\theta}_{t-1}) - \Phi(\theta_{t-1}, \pi_{t-1}) + \frac{L}{2}\|\hat{\theta}_{t-1} - \theta_{t-1}\|^2$$
$$\leq F(\theta_{t-1}) - \Phi(\theta_{t-1}, \pi_{t-1}) - \frac{L}{2}\|\hat{\theta}_{t-1} - \theta_{t-1}\|^2$$

The first line follows by convexity and $L-$smoothness. The second line by definition of $F$. The third line by definition of $\hat{\theta}$. Next, by definition of Moreau envelop,

$$F_{1/2L}(\theta_t) \leq F(\hat{\theta}_{t-1}) + L\|\hat{\theta}_{t-1} - \theta_t\|^2$$

Take expectation on both sides and we get

$$\mathbb{E}[F_{1/2L}(\theta_t)] \tag{11}$$

$$\leq F(\hat{\theta}_{t-1}) + \mathbb{E}[L\|\hat{\theta}_{t-1} - \theta_t\|^2]$$

$$= F(\hat{\theta}_{t-1}) + L(\|\theta_{t-1} - \hat{\theta}_{t-1}\|^2 + 2\eta_\theta\langle\hat{\theta}_{t-1} - \theta_{t-1}, \nabla_\theta\Phi(\theta_{t-1}, \pi_{t-1})\rangle + \eta_\theta^2(l^2 + \sigma^2))$$

$$\leq F_{1/2L}(\theta_{t-1}) + 2L\eta_\theta(\Phi(\hat{\theta}_{t-1}, \pi_{t-1}) - \Phi(\theta_{t-1}, \pi_{t-1}) - \frac{L}{2}\|\hat{\theta}_{t-1} - \theta_{t-1}\|^2) + L\eta_\theta^2(l^2 + \sigma^2)$$

The second line substitutes in (9). The third line follows by convexity and $L-$smoothness. Denote that $\Delta_t := F(\hat{\theta}_{t-1}) - \Phi(\theta_{t-1}, \pi_{t-1}) \geq \Phi(\hat{\theta}_{t-1}, \pi_{t-1}) - \Phi(\theta_{t-1}, \pi_{t-1})$. We can sum over $t$ and take expectation recursively to get,

$$\sum_t \mathbb{E}[\|\nabla F_{1/2L}(\theta_t)\|^2] = 2L\sum_t \mathbb{E}[\|\hat{\theta}_{t-1} - \theta_{t-1}\|^2] \tag{12}$$

$$\leq \frac{2}{L\eta_\theta}(F_{1/2L}(\theta_0) - F_{1/2L}^*) + 4\sum_t \Delta_t + T\eta_\theta(l^2 + \sigma^2))$$

where $F_{1/2L}^* = \min_\theta F_{1/2L}(\theta)$. The first equality follows by the definition of Moreau envelope. The second inequality follows by rearranging (11).

Next, we aim to bound the accumulated error $\sum_t \Delta_t$.

Recall that the update for the $\pi$ is as follows for $\mathbb{E}[g_\pi] = \nabla_\pi f(\theta, \pi)$,

$$\pi_{k+1} := \text{argmin}_{\pi\in\Delta^L}\{-2\lambda\langle g_\pi, \pi\rangle - 2\lambda h(\pi) + \text{KL}(\pi, \pi_k)\}$$

Applying Lemma 5 with $L(\pi) = -2\lambda\langle g_\pi, \pi\rangle + -2\lambda h(\pi)$, we get

$$-h(\pi^*(\theta_s)) - \langle g_\pi, \pi^*(\theta_s)\rangle + \text{KL}(\pi^*, \pi_k)$$
$$\geq -\langle g_\pi, \pi_{k+1}\rangle - h(\pi_{k+1}) + \text{KL}(\pi_{k+1}, \pi_k) + \text{KL}(\pi^*, \pi_{k+1})$$

Rearrange and take expectation we get

$$2\lambda(\mathbb{E}[\langle -g_\pi, \pi_k - \pi^*(\theta_s)\rangle] - \mathbb{E}[h(\pi_{k+1})] + \mathbb{E}[h(\pi^*(\theta_s))])$$

$$\leq -\mathbb{E}[\text{KL}(\pi_{k+1}, \pi_k)] + \text{KL}(\pi^*, \pi_{k+1}) - \text{KL}(\pi^*, \pi_k) + 2\lambda(\mathbb{E}[\langle g_\pi, \pi_k - \pi_{k+1}\rangle])$$

$$\leq -\mathbb{E}[\text{KL}(\pi_k, \pi^*)] + \mathbb{E}[\text{KL}(\pi^*, \pi_{k+1})] - \|\pi^* - \pi_k\|_1^2/2 + 2\lambda^2\mathbb{E}[\|g_\pi\|_\infty^2] + \|\pi^* - \pi_k\|_1^2/2$$

The second inequality follows by the fact that $KL-$divergence is strongly convex with respect to $L_1$ norm and Cauchy-Schwartz inequality. We further observe that

$$-\mathbb{E}[\langle g_\pi, \pi_k - \pi^*(\theta_s)\rangle] = -\langle\nabla_\pi f(\theta_k, \pi_k), \pi_k - \pi^*(\theta_s)\rangle \geq -f(\theta_k, \pi_k) + f(\theta_k, \pi^*(\theta_s))$$
$$= -f(\theta_k, \pi_k) + f(\theta_k, \pi^*(\theta_k)) - f(\theta_k, \pi^*(\theta_k)) + f(\theta_k, \pi^*(\theta_s))$$
$$\geq -f(\theta_k, \pi_k) + f(\theta_k, \pi^*(\theta_k)) - f(\theta_k, \pi^*(\theta_k)) + f(\theta_s, \pi^*(\theta_k))$$
$$\quad - f(\theta_s, \pi^*(\theta_s)) + f(\theta_k, \pi^*(\theta_s))$$
$$\geq -f(\theta_k, \pi_k) + f(\theta_k, \pi^*(\theta_k)) - 2l\|\theta_s - \theta_k\|$$

The first inequality follows by concavity. The third line follows by $f(\theta_s, \pi^*(\theta_k)) \leq f(\theta_s, \pi^*(\theta_s))$. The last inequality follows by Lipschitzness. Similarly,

$$-h(\theta_k, \pi_k) + h(\theta_k, \pi^*(\theta_s)) \geq -h(\theta_k, \pi_k) + h(\theta_k, \pi^*(\theta_k)) - 2l\|\theta_s - \theta_k\|$$

We can take iterative expectation and get sum over $k = s + 1, ..., s + B$ to get

$$\sum_{k=s+1}^{s+B} \mathbb{E}[-f(\theta_k, \pi_k) + f(\theta_k, \pi^*(\theta_k)) - h(\pi_k) + h(\pi^*(\theta_k))]$$

$$\leq -h(\pi_s) + \mathbb{E}[h(\pi_{s+B})] + 4l\sum_{k=s+1}^{s+B}\sum_{j=s}^{k}\mathbb{E}[\|\theta_j - \theta_{j+1}\|] + \lambda BG^2$$

$$+ \frac{1}{2\lambda}(\mathbb{E}[\text{KL}(\pi^*(\theta_s), \pi_0)] - \mathbb{E}[\text{KL}(\pi^*(\theta_s), \pi_s)])$$

Note that $\Delta_k = -f(\theta_k, \pi_k) + f(\theta_k, \pi^*(\theta_k)) - h(\pi_k) + h(\pi^*(\theta_k))$, hence

$$\sum_{k=s+1}^{s+B} \mathbb{E}[\Delta_k] \leq -h(\pi_s) + \mathbb{E}[h(\pi_{s+B})] + 2\eta_\theta l B^2 G + \lambda B G^2$$

$$+ \frac{1}{2\lambda}(\mathbb{E}[\mathrm{KL}(\pi^*(\theta_s), \pi_0)] - \mathbb{E}[\mathrm{KL}(\pi^*(\theta_s), \pi_s)])$$

By further sum over all blocks and divide by total number of iterations $T$, we get

$$\frac{1}{T}\sum_{b=1}^{T/B}\sum_{k=bs+1}^{s+B} \mathbb{E}[\Delta_k] \leq (-h(\pi_0) + \mathbb{E}[h(\pi_T)])/T + 2\eta_\theta BG + \lambda G^2$$

$$+ \frac{1}{2\lambda B}(\mathbb{E}[\mathrm{KL}(\pi^*(\theta_s), \pi_0)] - \mathbb{E}[\mathrm{KL}(\pi^*(\theta_s), \pi_s)])$$

Substitute the above inequality into (12) and we get

$$\frac{1}{T}\sum_t \mathbb{E}[\|\nabla F_{1/2L}(\theta_t)\|^2] \leq \frac{2}{TL\eta_\theta}(F_{1/2L}(\theta_0) - F_{1/2L}^*) + 4(-h(\pi_0) + \mathbb{E}[h(\pi_T)])/T + 2\eta_\theta BG + \lambda G^2$$

$$+ \frac{1}{2\lambda B}(\mathbb{E}[\mathrm{KL}(\pi^*(\theta_s), \pi_0)] - \mathbb{E}[\mathrm{KL}(\pi^*(\theta_T), \pi_T)]) + \eta_\theta(l^2 + \sigma^2)^{1/2})$$

If we set $\eta_\theta = T^{-3/4}, B = T^{1/2}, \lambda = T^{-1/4}$, then we see that

$$\frac{1}{T}\sum_t \mathbb{E}[\|\nabla F_{1/2L}(\theta_t)\|^2] \leq \frac{1}{T^{1/4}}\left(\frac{2}{L}(F_{1/2L}(\theta_0) - F_{1/2L}^*) + 2G + G^2 + \frac{R}{2} + (l^2 + \sigma^2)^{1/2}\right)$$

$$+ (-h(\pi_0) + h^*)/T$$

$\square$

**Lemma 5.** *For any differentiable convex function $L$, if $x^* = \mathrm{argmin}_{x\in\Delta}\{L(x) + \mathrm{KL}(x, x_0)\}$, then for any $x' \in \Delta$, we have*

$$\ell(x') + \mathrm{KL}(x', x_0) \geq \ell(x^*) + \mathrm{KL}(x^*, x_0) + \mathrm{KL}(x', x^*).$$

*Proof.* This Lemma is well-known, but we include a proof for completeness. By optimality of $x^*$ and convexity of $\Delta$, we know that

$$\langle\nabla\ell(x^*) + \nabla\phi(x^*) - \nabla\phi(x_0), x - x^*\rangle \geq 0,$$

where $\phi(x) = \sum_i x_i \log(x_i)$, and the Bregman divergence defined according to $\phi$ is KL-divergence. Then

$$\begin{aligned}
\ell(x') &\geq \ell(x^*) + \langle\nabla\ell(x^*), x' - x^*\rangle \\
&\geq \ell(x^*) + \langle\nabla\phi(x_0) - \nabla\phi(x^*), x - x^*\rangle \\
&= \ell(x^*) - \langle\nabla\phi(x_0), x^* - x_0\rangle + \phi(x^*) - \phi(x_0) \\
&\quad + \langle\nabla\phi(x_0), x' - x_0\rangle + \phi(x') - \phi(x_0) - \langle\nabla\phi(x^*), x' - x^*\rangle + \phi(x') - \phi(x^*) \\
&= \ell(x^*) + \mathrm{KL}(x^*, x_0) - \mathrm{KL}(x', x_0) + \mathrm{KL}(x', x^*)
\end{aligned}$$

$\square$

