# OpenReview forum: "Coping with Label Shift via Distributionally Robust Optimisation"
_ICLR.cc/2021/Conference — ICLR 2021 Poster_

### Official Review · AnonReviewer4 · 2020-10-28

**Rating:** 6
**Confidence:** 4

**Review:**

Weighting each class differently during model training is a common technique to deal with label imbalance and label shift. In this paper, the authors propose AdvShift, a method for learning these weights in a data- and model-dependent way through distributionally robust optimization (DRO).

I found the paper especially well-written and clear, from the problem setup and contextualization within related work, to the description and intuition behind the method and the results, as well as the candid discussion of potential limitations of the method. Thank you to the authors for the enjoyable read.

Overall, the method and results seemed promising. However, I had two general areas of concern that prevented me from giving a higher rating; the first with the comparison to prior work, and the second about the experimental details.

**Prior work**
The authors identify Mohri et al. (2019) and Sagawa et al. (2020) as their closest work, so I primarily read up on and referred to those papers.

[1a] Compared to Mohri et al. (2019), the differences here seems to be that (a) Mohri et al. use a chi-squared regularizer vs. the KL divergence term used in AdvShift; (b) Mohri et al. use (computationally expensive) projections whereas AdvShift uses proximal updates; and (c) AdvShift also includes a couple of optimization tricks, loss clipping and gradient stabilization, as detailed in the appendix. The argument that the authors make is that (b) is the important ingredient; that is, it is a question of optimization. Is this a fair characterization?

[1b] If so, the tricks used in (c) seem to be critical to the performance of AdvShift, so they might perhaps explain the gap between Mohri et al. and AdvShift. The authors write that the proximal mirror ascent updates solve the optimization instability (compared to the projection operator), but do not elaborate.

[1c] Moreover, if the superiority of the method hinges on optimization, then the phenomenon discussed in the last paragraph is especially striking: it seems like AdvShift is not optimizing properly with regard to the algorithm threshold r, as the authors point out. This suggests that there could be fundamentally something wrong with the algorithm and its optimization.

[2] Compared to Sagawa et al. (2020), the authors write that the method in Sagawa et al. requires the ability to sample data from a given group. This does not appear to be true: Sagawa et al. describe their method as using standard minibatching and SGD. The authors also write that Sagawa et al. only test their method on a small number of labels, which is true, so it seems plausible that AdvShift will perform better on a large number of labels, but this would need to be empirically determined.

[3] Both Mohri et al. and Sagawa et al.’s methods were originally tested on a small number of labels. How does AdvShift compare to these when there are a small number of labels? In the Appendix, it appears that Mohri et al. performs comparably to AdvShift on CIFAR-100. Is the pitch that one should use AdvShift when there are a large number of labels, but that it does not matter otherwise?

**Experimental details**
[4] Fig 2(a) and 2(c): Why is there no difference in training error at tau = 0 even when training for a very conservative model (e.g., AdvShift 1.0, or Fixed 3)? This would make sense if train errors are close to 0 but they are not: Fig 6 in the appendix shows that the training errors are >20% at tau = 0. The models used seem very underparameterized, presumably as a result of hyperparameter tuning; do they actually beat a normal ResNet with standard hyperparameters in the adversarial setting? I couldn’t find hyperparameter tuning details in the appendix.

[5] Figs 2 and 3: Are the authors reporting test or validation accuracies? The figure captions all say validation, but the text all says test.

[6] Fig 2(d): I am confused by why the fixed baseline does so badly. To clarify, for each value of tau, does each model have its own adversarial shift? Specifically, for the fixed model, is it still being evaluated against the fixed distribution? If not, could we plot it as an oracle? If it is, that seems very strange to me that optimizing for exactly that distribution would get higher train error. The authors cite Byrd and Lipton (2019) but I wasn’t able to figure out how that paper addressed this case, since Byrd and Lipton seem to be operating in the separable data setting. This suggests that there are some optimization issues?

**Minor**
[7] It is reasonable to focus on the adversarial setting, but the adversarial test error is still exceedingly high despite some improvements from AdvShift, so from the point of view of providing guarantees, this is only marginally better. I’m curious if AdvShift helps on non-worst-case settings. Other work on label imbalance, for example, https://arxiv.org/abs/1901.05555 show that their reweighting techniques can improve accuracy on the standard test set for ImageNet.

Typos:
S4.1, “has a largely balanced training label distributions”
S4.2, “compre”

**Update**
Thank you to the authors for the detailed response. Overall, I'll leave my rating as a 6. The paper is clear and proposes a promising optimization method, with moderate comparisons to prior approaches and reasonable results on ImageNet (but not CIFAR-100). The impact of these results are somewhat diluted by the overall absolute low scores in the adversarial setting, but as a general DRO optimization method, the result is interesting. Thank you for all the discussion.

Comments to the latest author response (posting this as an edit since public comments are now disabled):

[1] Thanks, the updated version is clearer.

[2] I agree with this characterization (for example, the unconstrained perturbation might cause optimization instability). The paper still seems to contain the language about the Sagawa paper requiring a custom sampler. Overall, I agree that there are reasons to believe that the proposed optimization method is more stable, but I think the authors should be clear that they only compared to Mohri et al. experimentally (in your response to Reviewer 3, you mentioned that you extensively discussed and compared to both Mohri and Sagawa).

[6] Thanks. I also agree with this. I had two points: first, the differences between your proposed algorithm and the standard algorithm might be more stark when the training is not balanced. Second, in imbalanced settings, it is more likely that the test distribution will be skewed (e.g., rare animals in iNaturalist).

---

> ### Author Response · Authors · 2020-11-17
> **Thank you for the comments.**
>
> We thank the reviewer for the helpful comments. We are glad that the reviewer found the paper enjoyable to read.
>
> ***[1a] That is, it is a question of optimization. Is this a fair characterization?***
> Yes, the characterization is correct. Our main focus is to propose a practical and fast optimization algorithm to solve DRO. Regarding the specific sub-points:
> (a) Mohri et al.'s method can be applied to both Chi-squared divergence and KL divergence; however, see comment below regarding our use of KL divergence.
> (b) the projection method would be computationally expensive (see Appendix E), whereas we circumvent this via Lagrangian dual formulation. Note here that the use of KL divergence allows for closed-form updates.
> (c) other tricks (e.g., gradient clipping) provide additional improvements (see Figure 7, 8 in the Appendix for ablation studies).
>
> ***[1b] The authors write that the proximal mirror ascent updates solve the optimization instability but do not elaborate.***
> At a high level, mirror descent can achieve better dependence on the variable dimension (in our setting, the number of classes) compared to SGD. If we reduce the stochastic optimization problem as a special case of online learning, this better dependence means that mirror descent can tolerate higher noise while achieving similar convergence rate. Such improvement is possible because mirror descent exploits the simplex structure of the decision variable. More details in [Sebastien Bubeck. Convex optimization: Algorithms and complexity]
> The proximal method is helpful when the objective can be decomposed into two parts, and one of the two (in our case, the KL regularization) can be solved in closed form. This speeds up convergence by removing additional computation complexity caused by solving the regularization objective (as it usually leads to higher Lipschitz constant and second moment of gradient) from solving both part altogether. More details in [Patrick L Combettes and Jean-Christophe Pesquet. Proximal splitting methods in signal processing.]
> In summary, both methods exploit the problem structure to speed up convergence when the noise level is controlled.
>
> ***[1c] AdvShift is not optimizing properly with regard to the algorithm threshold r.***
> We agree with the reviewer that ideally, the algorithm should optimize against the prefixed threshold r. However, in practice a larger radius usually leads to a more difficult problem due to increased noise. Compared with the agnostic method, we can see that our proposed method significantly improved optimization performance with large perturbation distance.
>
> ***[2] Sagawa et al. describe their method as using standard minibatching and SGD.***
> The authors use minibatch SGD, but altered the sampling procedure for getting minibatches, where they assume all samples within the same batch comes from the same subgroup/label class. For the Imagenet data pipeline, sampling from a single label can lead to a computational overhead.
>
> ***[3] Is the pitch that one should use AdvShift when there are a large number of labels?***
> This is a fair characterization. We emphasize that the challenging large label setting is more reflective of what one often encounters in practice.
>
> ***[4] Why is there no difference in training error at tau = 0?***
> Our hypothesis is that if the network can be successfully optimized, then it can have enough capacity to fit much more samples than when trained with standard ERM. Therefore, by improving the performance on hard classes without losing much on easy ones, it could be robust against shift while preserving the performance on a balanced validation set.
>
> ***[4] The models used seem very underparameterized; do they actually beat a normal ResNet with standard hyperparameters***
> We used the standard ResNet50 architecture as in the original paper. The validation accuracy is 76%, which matches what was originally reported (see the official repo https://github.com/KaimingHe/deep-residual-networks).
>
> ***[5] Test or validation accuracies?***
> We are reporting performance on the standard ImageNet validation set. We have updated the text accordingly.
>
> ***[6] I am confused by why the fixed baseline does so badly.***
> For each model and τ value, there is an adversarial distribution: this is the distribution that maximizes the model error rate given a perturbation threshold τ. Consequently, when training a model using a fixed label distribution (as in most prior work on label distribution shift), during evaluation, the model would have an adversarial distribution that is *different* from the fixed one during training. Hence the performance is expected to be suboptimal. This shows that up weighting the hard classes (as measured by validation error of a baseline ResNet50 model) does not lead to a robust model, because the model now is prone to errors in other classes.
>
> ***[7]*** We thank the reviewer for this suggestion. We agree that this would be an interesting direction for future research.

---

> > ### Comment · AnonReviewer4 · 2020-11-24
> > **Response**
> >
> > Thank you to the authors for the helpful response. I agree on all points, except that in my opinion, the baseline comparisons are still incomplete. Specifically, keeping the original numbering:
> >
> > [1b] As mentioned above, the gradient clipping and stabilization techniques, which were relegated to the appendix, seem to have a very large effect on performance. In my opinion, I think it would be helpful to clarify that the main contribution of this paper is in optimization, especially since the authors state that the method of Mohri et al. optimizes for the same worst-case objective but fails because of optimization stability, and to promote these "tricks" to a discussion in the main text.
> >
> > [2] With all due respect, based on my reading of Sagawa et al. and skimming their associated source code, I believe that the authors are incorrect here. The sampling procedure they use for getting minibatches appears to be standard.
> >
> > I readily agree that there are multiple reasons why the authors' proposed method might be better than these baselines. For example, they show that it works on a large number of labels (whereas the original papers for these baselines did not), they argue that the method of Mohri et al. is computationally slow, etc. But I think the onus is on the authors to directly show these results.
> >
> > [6] Regarding the fixed baseline: Thanks. That makes sense. As an oracle / upper bound on potential accuracy, it would have been helpful to see how well the fixed baseline does against the original adversarial distribution.
> >
> > As an aside, I wanted to note that the original forms of CIFAR-100 and ImageNet are not often used to test label shift (because as the authors note, their label distributions are already quite balanced, whereas there are many other real-world datasets where label shift is an issue). One might expect that on other datasets with imbalanced labels, the gap between standard methods and the proposed method would be even larger.
> >
> > Overall, the technical method seems solid, but I still think that the paper would benefit from clarifying the optimization contribution and its performance and differences relative to baselines/oracles, and perhaps on datasets where label imbalance and shift is more of an issue.

---

> > > ### Author Response · Authors · 2020-11-24
> > > **Thank you for the discussion**
> > >
> > > We thank the reviewer for the prompt feedback and suggestions. We updated the draft and address the points as follows
> > >
> > > **[1b]  I think it would be helpful to clarify that the main contribution of this paper is in optimization**
> > > - We updated our main contribution to emphasize the optimization aspect.
> > > - We promoted the discussion about clipping and epsilon to main text. Please also note that even without these two tricks, the improvement is significant. We don't have the unclipped experiment now, but clipping at loss 10 is roughly unclipped given that a uniform noisy prediction will give loss = log(1000) ~= 6.9. Hence, as we explained, both our proposed algorithm and the clipping/epsilon procedure improves the convergence independently. The combination leads to the result reported in the experiment section.
> > >
> > >
> > > **[2]The sampling procedure they use for getting minibatches appears to be standard. They argue that the method of Mohri et al. is computationally slow, etc. But I think the onus is on the authors to directly show these results.**
> > > - In Alg 1 of the Sagawa paper, the line 2 within the for loop assume all x, y are from the same subgroup.
> > > - We also checked the code, and it appears importance sampling is used in loss.py file, so the pipeline in code is standard as the reviewer mentioned. However, this only makes the algorithm closer to the agnostic method, where instead of doing an gradient ascent, Sagawa's method does exponentiated update.
> > > - The sampling issue is just one problem, which can be easily fixed with importance sampling. The main challenge is on how to stabilize the additional noise incurred by doing importance sampling and how to constrain the perturbation radius without doing the expensive projection step.
> > > - Please refer to Appendix E for an empirical study of the projection step.
> > >
> > >
> > > **[6]  CIFAR-100 and ImageNet are not often used to test label shift.**
> > >
> > > We hope to clarify why we did not test our method on imbalanced data such as iNaturalist. In brief, the setting where there is a fixed target data that is different to the source data is not the setting we want to address. In detail:
> > > - Previous work on label shift assumes access to the target data, so that either the target label distribution is known (e.g. balanced), or can be estimated by evaluating the samples with a trained model. With the latter, there is a computational overhead due to additional estimation procedure after training the baseline model. More importantly, however, the output model is robust **only** to the target data.
> > > - In contrast, our proposed model does not have access to the target data, and is robust to **any** target distribution suitably close to the training distribution. Such robustness typically involves a tradeoff: the method may perform inferior to other label shift baselines can directly estimate (or know) the target label distribution.
> > > - If, however, we make all algorithms agnostic to the target data, then it does not matter whether the training is balanced or not: we only need to adversarially perturb the label distribution. Hence, CIFAR and ImageNet would suffice for this purpose.
> > >
> > > In summary, we very much appreciate the reviewer for the careful and detailed comments as they help improve our draft. We are also happy to include additional edits/experiments in final version after the discussion is over.

---

### Official Review · AnonReviewer3 · 2020-10-28
**Good submission but lacks some coherency**

**Rating:** 4
**Confidence:** 4

**Review:**

This paper tackles label shift in supervised learning via distributionally robust optimization. The main idea is to train by solving a min-max problem, where the max problem searches for the worst-case label shift in an Kullback-Leibler divergence ambiguity set. The KL ambiguity set will generate some form of adversarial reweighing of the sample training points, which gives us hope that the learned parameters will perform better in the test data (with shift). The paper proposes to solve the Lagrangian version instead of the constrained version of the DRO problem, and proposes a gradient descent-ascent type of algorithm.

The main contributions of this paper is condensed in Section 3:
- From the mathematical viewpoint, the DRO problem (3) and the mini batch gradient estimator are not new.
- In Section 3.2, the authors proposed the Lagrangian problem. However, the difficulty of this problem lies in choosing the "right" value of gamma_c. Indeed, finding gamma_c is not trivial, and simply saying that a bisection search can be employed (as suggested by the authors) is not sufficient. To use bisection search, the authors should argue that the optimal value of problem (4) is at least convex in gamma_c, and this result has not been established.
- The authors relax the 2-norm to a KL-divergence in equation (6). This relaxation makes the complaint (2) in page 4 become invalid. Indeed, complaint (2) says that projection is difficult under 2-norm, however, we can also relax to the projection using KL-divergence to alleviate this difficulty.
- Problem (3) and (4) are equivalent only if gamma_c is optimally tuned. The algorithm analyzed in Section 3.4 is has guaranteed only for problem (4). It is still not clear what guarantee we can have on the original problem (3), which is the main problem of interest.

Getting back to the main contributions that are listed in page 2: I can only partially agree with contribution (2). However, I do not think that contribution (1) about the formulation and contribution (3) about the numerical results can really be justified as the main contributions of this paper.

Minor comment:
1. In algorithm 1, line 1 should be \pi_0. In line 2, the for loop should start with t = 0
2. I don't understand why p_emp should not be on the boundary of the simplex. If p_emp is on the boundary, we can simply drop the samples (x_i, y_i) with p_emp(y_i) = 0, and that should not affect the problem.

---

> ### Author Response · Authors · 2020-11-17
> **Thank you for the comments.**
>
>
> We thank the reviewer for the helpful comments and address them as follows.
>
> ***The DRO problem (3) and the mini batch gradient estimator are not new... I do not think that contribution (1) about the formulation and contribution (3) about the numerical results can really be justified as the main contributions of this paper***
>
> One may consider it natural to solve a DRO problem in order to find a robust model against label shifts. However, carefully designing the optimizer to ensure the *practical effectiveness* of such a technique is a novel contribution. Indeed, despite the large body of work on label shift, the DRO approach was never taken (see Table 1). Further, to our knowledge, no previous procedure can perform end-to-end DRO training with neural networks at ImageNet scale. In particular:
>     - [1] gives novel result on the generalization behavior, but its experiments are solved with convex optimizers, which does not scale with data size and variable dimension.
>     - the only applicable algorithms we found are [2], [3] which we extensively discussed and compared. While these algorithms have convergence guarantees, they are not tailored for problems when the number of classes/subgroups is large (such as Imagenet).
>
> To speed up convergence, there are a plethora of techniques that one could potentially employ (e.g., variance reduction, proximal/mirror descent, debiasing, projection, momentum, adaptivity, ...). However, given that each of these techniques makes assumptions not satisfied by neural network training, it is unclear which of these actually help in the label shift scenario. In our work, we propose the first setup that is shown to be effective on ImageNet scale.
>
>
> ***However, the difficulty of this problem lies in choosing the "right" value of gamma_c. Indeed, finding gamma_c is not trivial, and simply saying that a bisection search can be employed (as suggested by the authors) is not sufficient... Problem (3) and (4) are equivalent only if gamma_c is optimally tuned.***
>
> Please note that in Proposition 1, we only need to find a γ_c larger than a certain threshold, rather than finding an exact optimal value. Intuitively, γ_c needs to be large enough such that penalty incurred by violating the constraint is higher than the gain in minimizing the objective. The level of robustness is controlled by r instead of γ_c.
>
> In practice, setting a very large γ_c may lead to convergence issues as it generates large gradients for the adversarial distribution π. Therefore, we employ a bisection search to find a good γ_c such that it is large enough to constrain the adversary, but small enough to make the adversarial update stable.
>
>
> ***We can also relax to the projection using KL-divergence to alleviate this difficulty.***
>
> Thanks for pointing out the lack of clarity on this point. We added a section in Appendix D to explain the difference between optimizing an objective regularized by a KL divergence and optimizing an objective constrained by a KL divergence.
>
> In short, even for the KL-divergence, the constrained version (which is required by projection operator) does not permit a Boltzmann distribution as an optimal solution, whereas the regularized version does. The intuition is that to find the optimal solution, one can set the gradient equal 0 for the regularized version; however, for the constrained version, one would need to solve the KKT conditions. For exactly this reason, we use the Lagrangian dual to iteratively solve regularized versions as a surrogate loss.
>
>
> ***I don't understand why p_emp should not be on the boundary of the simplex***
>
> When p_emp is on the boundary, the smoothness constant of the KL divergence becomes unbounded. This is also an unrealisitic situation, where one of the class has zero samples.
>
>
> [1] Hongseok Namkoong, John C. Duchi, Variance-based Regularization with Convex Objectives. Neurips
> [2] S. Sagawa, P. W. Koh, T. B. Hashimoto, and P. Liang, Distributionally robust neural networks forgroup shifts: On the importance of regularization for worst-case generalization. ICLR
> [3] Mehryar Mohri, Gary Sivek, and Ananda Theertha Suresh. Agnostic federated learning. ICML

---

### Official Review · AnonReviewer1 · 2020-10-29
**Coping with Label Shift via Distributionally Robust Optimisation**

**Rating:** 7
**Confidence:** 4

**Review:**

This paper attacks the issue of mismatch in distribution of labels between train and test samples. The authors propose a DRO-based approach which amounts to solving a modified ERM problem. Compared to classical approaches, the proposed method doesn’t entail fitting many different models: just a single model. The method builds on recent progress on solving nonconvex-concave games for approximate stationary points. The resulting algorithm is a mirror-descent scheme with explicit convergence rates, under Setting : train and test label distribution do not match

The general approach is:
Learn a classifier robust to arbitrary label shifts (from a family) -> doing this by using DRO approach because it allows them to train a model that performs well on all label distributions sufficiently close to the training data label distribution.
The paper proposes the DRO approach with KL-divergence for the label shift problem, an analysis of the proposed algorithm (optimization techniques and convergence analysis) and experiments on Imagenet (ResNet50)

Good points
      - It paper is well-written and asy to follow paper. The problem and the algorithm are clearly described and well analyzed. The experiments are also quite well done.
The label shift pb is clearly explained and described
The analysis of the proposed algorithm (more or less all section 3) is quite thorough.
The experiment section is clearly described (and the additional experiments in Appendix are useful).

Bad points:
The experiments are not really convincing in the sense that the improvement is either really limited on the test set by comparisons to the baseline (on Imagenet), or similar to that of the Agnostic method (on CIFAR-100, Appendix)
It would have been nice to have a more detailed explanation of the complexity of the algorithm in practice, and more explanation on the choices of hyper-parameters (Appendix B). There are many hyper-parameters that were set to a fixed value, and only one parameter is tuned (how?), so how/why do you choose these fixed values?

Recommendations:
      -  Contrary to what the authors say, inner problem in (4) does indeed have a closed-form solution. It is the Fenchel-Legendre transform of KL divergence, which equals log-sum-exp(...). Also, this max is attained at Boltzman distribution which is proportional to exp(loss / temperature). See Lemma 4 of Faury et al. (AAAI 2020) Distributionally Robust Counterfactual Risk Minimization. BTW, its would be interesting to contrast this result with the result of Lemma 2, I. Equation (7),  of the current manuscript.

Errors:
      -  Equation (2) and previous equation (unlabeled): yi should be replaced with y
      - First equation  (unlabeled) in section 3.1: missing pi(y) in importance weight

Questions
It is mentioned that the Agnostic method faces many challenges during training, but these challenges are not explained afterwards (computation time? Difficult parameters tuning? Etc.)

I’ll happily revise my score upward if my concerns are addressed.

---

> ### Author Response · Authors · 2020-11-17
> **Thank you for the comments.**
>
> We thank the reviewer for the helpful comments. We particularly appreciate the reviewer's comments that the paper is well written and the algorithm is well analyzed.
>
>
>
> ***The improvement is either really limited on the test set by comparisons to the baseline (on Imagenet), or similar to that of the Agnostic method (on CIFAR-100, Appendix)***
>
> 1. Our method improves upon the baseline on the validation set by ~3%. Such an improvement without any modification to architecture or additional computation time is non-trivial. For example, the batch norm technique improved upon the Inception model by ~2.5% in its first publication.
>
> 2. Our method has even bigger gains over the baseline on the training set (~8%). We found that standard techniques (e.g., weight decay) do not significantly mitigate this generalization gap. Studying how to further reduce this generalization gap between train and validation performance would be an interesting topic for future work.
>
>
>
> ***More detailed explanation of the complexity of the algorithm in practice***
>
> Per point (2) on pg 4, we designed AdvShift to have minimal computational overhead compared to baselines (e.g., SGD). In particular, we use Lagrangian duality instead of projected SGD, which avoids an additional ~1 min per iteration to solve the projection (see Appendix E). Consequently, the majority of computation time is spent on neural network inference and backpropagation, and our algorithm modifies SGD only by a few closed form updates on the adversary distribution. In practice, AdvShift incurs <5% overhead compared to standard SGD.
>
>
>
> ***More explanation on the choices of hyper-parameters. How/why do you choose these fixed values?***
>
> Per Appendix B, we picked the clipping parameter and ε value based on ablation studies; see Figure 7, 8. We also found the performance to be insensitive to the choice of 2 γ_c λ, and hence set that to be 1. In the body, we fixed all the above hyperparameters and present results for various adversarial distribution learning rate, as well as the robustness threshold r.
>
>
>
> ***Inner problem in (4) does indeed have a closed-form solution***
>
> Thanks for referring us to this interesting result. In Lemma 4 of the reference, the claim is that the maximum of (4) can be achieved for some Boltzmann distribution on an unregularized objective. However, this distribution may not be the solution to the original DRO. In other words, the *function value* matches, but the *solution* does not. Please see the updated Appendix D for a concrete example.
>
> Note that in order to solve the min-max problem, we need to find the (approximate) optimal solution to the inner problem of (4) instead of finding its function value. Therefore, we take the Lagrangian dual approach.
>
>
>
> ***It is mentioned that the Agnostic method faces many challenges during training, but these challenges are not explained afterwards (computation time? Difficult parameters tuning? Etc.)***
>
> The difficulty arises from both parameter tuning as well as computation time. In detail:
>
> - we found that when training ImageNet with the Agnostic method, no choice of learning rate leads to convergence within 90 epochs (see Figure 2). In short, the Agnostic method fails to optimize the DRO in the ImageNet setting.
>
> - one possible reason for Agnostic's training instability is that it does not limit the perturbation radius of the adversarial distribution. As one has to use importance sampling to get an unbiased gradient estimate, allowing the adversarial to perturb arbitrarily can lead to extreme ratios in the reweighting step within importance sampling.
>
> - one may constrain the adversary to overcome the above and stabilize training. One natural means of doing so is via a projection step. However, this incurs a non-trivial computation overhead (see Appendix E).

---

> > ### Comment · AnonReviewer1 · 2020-11-24
> > **My final comment**
> >
> > Thanks to the authors for their response. I'll keep my previous score of 6. I think the presentation of the contributions of the paper could be made more sharp. Tighter links to existing literature should be made (see my previous comment). Also, the specifics / regime of the scenario considered in the paper should be outlined very early in the paper.

---

> > > ### Author Response · Authors · 2020-11-25
> > > **Thank you for the suggestion.**
> > >
> > > We have made comments in the main text about the literature and added the reference.
> > >
> > > In the main text, most background and setup were along the label shift literature, which is our problem of interest. More literature on DRO are discussed later in Appendix due to space limit.
> > >
> > > We also appreciate the reviewer for considering our work as well written and easy to follow. We are willing to make further edits if the reviewer could provide more detail on what additional specifics need to be specified.

---

> > > > ### Comment · AnonReviewer1 · 2020-11-25
> > > > **Final comments**
> > > >
> > > > OK, thanks for the update. The situation looks much clearer now. I've increased my previous score increased to 7.
> > > >  I recommend accepting this paper for ICLR 2021.

---

### Decision · Program_Chairs · 2021-01-07
**Final Decision**

**Decision:**

Accept (Poster)

**Comment:**


This paper considers the problem of training neural networks to be robust to label shifts. To do so, it proposes to use a distributionally robust optimization (DRO): instead of minimizing the expected error with respect to the empirical data distribution, the worse case expected error is minimized over a KL-divergence "ball" of distributions centered at the empirical distribution with a given radius. The main contribution of the paper is an efficient algorithm for achieving this optimization, that avoids the need to project onto the uncertainty set or to sample in non-standard ways from the training set. The paper provides evidence on the ImageNet data set and the ResNet-50 architecture that the proposed AdvShift algorithm outperforms reasonable baselines.

Reviewers raised concerns about the novelty of the algorithm, and claims the paper makes about the infeasibility of the sampling required by one of the competing baselines, and the need for the Lagrangian parameter used in the algorithm to be well-tuned. The rebuttals addresses the concerns suitably; in particular, the novelty of the algorithm lies it it being the first DRO-based solution for the label shift problem and its efficiency obtained by using KL uncertainty set and the Lagrangian formulation of the problem, which allows a closed form solution.

Due to the strong empirical and theoretical performance of the proposed AdvShift algorithm, it is recommended that this paper be accepted.